# Characterizing the Alteration in Rumen Microbiome and Carbohydrate-Active Enzymes Profile with Forage of Muskoxen Rumen through Comparative Metatranscriptomics

**DOI:** 10.3390/microorganisms10010071

**Published:** 2021-12-30

**Authors:** Xiaofeng Wu, Chijioke O. Elekwachi, Shiping Bai, Yuheng Luo, Keying Zhang, Robert J. Forster

**Affiliations:** 1Lethbridge Research and Development Centre, Agriculture and Agri-Food Canada, Ottawa, ON K1A 0C6, Canada; wxfeng502@163.com (X.W.); c_elekwachi@yahoo.com (C.O.E.); 2Institute of Animal Nutrition, Sichuan Agricultural University, Ya’an 625014, China; shipingbai@sicau.edu.cn (S.B.); luoluo212@126.com (Y.L.)

**Keywords:** muskoxen rumen, metatranscriptomics, microbiome, triticale straw, brome hay

## Abstract

Muskox (*Ovibos moschatus*), as the biggest herbivore in the High Arctic, has been enduring the austere arctic nutritional conditions and has evolved to ingest and digest scarce and high lignified forages to support the growth and reproduce, implying probably harbor a distinct microbial reservoir for the deconstruction of plant biomass. Therefore, metagenomics approach was applied to characterize the rumen microbial community and understand the alteration in rumen microbiome of muskoxen fed either triticale straw or brome hay. The difference in the structure of microbial communities including bacteria, archaea, fungi, and protozoa between the two forages was observed at the taxonomic level of genus. Further, although the highly abundant phylotypes in muskoxen rumen fed either triticale straw or brome hay were almost the same, the selective enrichment different phylotypes for fiber degrading, soluble substrates fermenting, electron and hydrogen scavenging through methanogenesis, acetogenesis, propionogenesis, and sulfur-reducing was also noticed. Specifically, triticale straw with higher content of fiber, cellulose selectively enriched more lignocellulolytic taxa and electron transferring taxa, while brome hay with higher nitrogen content selectively enriched more families and genera for degradable substrates-digesting. Intriguingly, the carbohydrate-active enzyme profile suggested an over representation and diversity of putative glycoside hydrolases (GHs) in the animals fed on triticale straw. The majority of the cellulases belonged to fiver GH families (i.e., GH5, GH6, GH9, GH45, and GH48) and were primarily synthesized by *Ruminococcus*, *Piromyces*, *Neocallimastix*, and *Fibrobacter*. Abundance of major genes coding for hemicellulose digestion was higher than cellulose mainly including GH8, GH10, GH16, GH26, and GH30, and these enzymes were produced by members of the genera *Fibrobacter*, *Ruminococcus*, and *Clostridium*. Oligosaccharides were mainly of the GH1, GH2, GH3, and GH31 types and were associated with the genera *Prevotella* and *Piromyces*. Our results strengthen metatranscriptomic evidence in support of the understanding of the microbial community and plant polysaccharide response to changes in the feed type and host animal. The study also establishes these specific microbial consortia procured from triticale straw group can be used further for efficient plant biomass hydrolysis.

## 1. Introduction

It is well established that the bioconversion of feed stuffs and energy outflow are synergistically conducted by various microbial populations dwelling in the rumen, including prokaryotes (bacteria and archaea) and eukaryotes (protozoa and fungi), with approximate 80% of the degradative activity contributed by bacteria and fungi, and 20% by protozoa [1]. Those microbial populations are featured by the high population density, vast diversity, diverse metabolic activities, and complexity of interactions [2,3,4], and mainly affected by feed types [5,6,7]. Many microbes have been individually characterized and functionally grouped into lignocellulolytic [8], electron transferring [9,10], proteolytic, amylolytic, lipolytic [11], and volatile fatty acid (VFA) producing subpopulation [12], which provided genetic information and fill the existing taxonomic information [9]. More important, these organisms produce a range of enzymes collectively known as glycoside hydrolase (GH) attributed for celluloses, hemicelluloses, and oligosaccharides deconstruction. Accordingly, rumen can be used as an excellent niche for exploring GH to be used in economic and efficient conversion of plant biomass to biofuels and other value-added products.

The symbiotic microbial communities residing inside the rumen are characteristic of both, the concerned animal species, and their diet. In the High Arctic, Muskoxen (*Ovibos moschatus*) as the largest herbivore has been enduring the austere arctic nutritional conditions and has evolved to ingest and digest scarce and high lignified forages to support the growth and reproduction [13,14]. Recently, there are more attentions focused on herbivores in the Arctic aiming at exploring the microorganism reservoir in their rumen [15,16,17,18], for seeking novel microbes or mechanisms may involve in the bioconversion of high fiber forages into microbial biomass, energy, or biofuel. For example, more lignocellulolytic enzymes in muskoxen rumen are transcript than in other well-known high fiber material fermenters such as macropod foregut, bovine rumen, and termite hindgut, and sequence annotation of putative protein encoding reads reveals that 63.8% of the sequences has been unknown and only small proportion of 14.4%, 6.8%, 3.4%, and 0.1% of sequences has been hit to fungi, protozoa, bacteria, and archaea, respectively [19]. Moreover, multiple-library comparisons between muskoxen and other ruminants have indicated that bacteria and archaea in the fecal sample of Norwegian muskoxen show more differences with those of other arctic ruminants and domesticated ruminants [18]. Dietary interventions and nature of the forage provided to the host have also been proposed to influence rumen microbiome for improved feed digestibility. For muskoxen, two high lignified forages of triticale straw and brome hay both as the main forage could be used to mimic the arctic forages and induce the metabolically active microbes due to their different in the fiber content. Collectively although studies have been conducted to investigate the microbiome of muskoxen, insights into the global microbial community and the effects of forage types on the rumen microbiome and glycoside hydrolase profile are still unavailable.

Application of small subunit (SSU) rRNA gene sequence analysis have demonstrated the inability of culture-dependent methods due to failed to discover the whole microbial structure and underestimate the diversity of the entire microbiota. Up to now, most of the investigations on the structure and composition of rumen microbiota have been conducted using SSU rRNA gene sequencing based on DNA-derived amplicons, such as prokaryotes [17,20,21], eukaryotes [22,23,24], and smaller taxonomic group [25]. This strategy could decipher the comprehensive diversity of primers-targeted groups including both the living and inactive microorganisms; however, it can’t reflect that the active microbial community changes corresponding to the environmental shifting. Although the modified strategy based on RNA-derived amplicons could describe the potential biological activity of the rumen microbial community in real time [26,27], it still fails to simultaneously investigate the prokaryotic and eukaryotic populations and inevitably introduce primer bias in PCR step [28,29]. In this context, metatranscriptomics could reveal the structure and diversity of active microbial community and metabolic pathways depending on the type of RNA, total RNA [30,31] or enriched mRNA [19,32,33] that used in sequencing library preparation. Given that the abundance of non-coding RNA accounts for approximately 99 % of total RNA, and the dominant comes from rRNA, metatranscriptomics based total RNA allows robust and simultaneous assessment of the metabolically active microbiome, with no need for prior selection of taxonomic groups for study. It also avoids PCR bias and can be straightforwardly carried out through a multiple sample and parallel sequencing approach [30]. In the view of the above, comparative metatranscriptomics was used for sequencing rumen metagenomes of muskoxen. The results enrich our information on the relationship between fiber digestion and microbial communities of livestock to improve the utility of rumen microbiome as a unique resource for mining diverse lignocellulolytic enzymes.

## 2. Materials and Methods

All the animals involving were cared in agreement with the protocol of by the Institutional Animal Care and Use Committee at the University of Alaska Fairbanks and fed at the Robert G. White Large Animal Research Station, Fairbanks, AK, USA. (No. 139821).

### 2.1. Experimental Design and Sample Collection

In a crossover design, 8 castrated muskoxen with ruminal cannulas were housed in individual pens and assigned into two groups based on their initial body mass (268 ± 18.4 kg vs. 278 ± 27.2 kg), and *ad libitum* fed with triticale (*Triticosecale hexaploide*) straw as a low-quality diet or brome (*Bromus* spp.) hay as a medium-quality diet from August to September (Appendix A). The chemical composition of which showed in Appendix A. Each stage contained three weeks for diet adaptation and one week for sampling and study. During the period, animals were offered the forages twice daily (mid-morning and late afternoon), plus a supplement of 335 g d-1 of protein and mineral (M Ration, Alaska Pet and Garden, Anchorage, AK; 2.24% nitrogen, 24% Neutral Detergent Fiber (NDF), 16% Acid Detergent Fiber (ADF), 2% lignin and 2% total ash, DM basis) once at mid-morning, and had continuous access to fresh water or snow. Due to an infection of the horn boss, one of the animals was removed from the experiment.

Before the morning feeding on the fourth day of sampling week, solid content was collected from different site of rumen and was filtered out the rumen fluid using a coffee filter plunger (Bodum Inc., Triengen, Switzerland). Subsamples of solid digests (about 5 g) were obtained according to Meng Qi’s method [19] and frozen in liquid nitrogen for further application.

### 2.2. RNA Extraction

To mostly efflux the total RNA, a modification of the method described by Wang [34] was used. Briefly, ruminal solids were manually ground to a fine powder in liquid nitrogen using a liquid-nitrogen pre-chilled mortar and pestle and performed for further 5 min grinding in liquid nitrogen using a Retsch RM100 grinder (Retsch GmbH, Haan, Germany). 1 g of each milled sample was weighed and 10 volumes of Trizol^®^ Reagent (Life Technologies, Carlsbad, CA, USA) was added into. RNA isolation was carried out according to protocol of Trizol^®^ Reagent. Small fragments and contaminants in the extracted total RNA samples were removed using MEGAclearTM kit (Life Technologies, Carlsbad, CA, USA) and OneStepTM PCR Inhibitor Removal Kit (Zymo Research, Irvine, CA, USA). The integrity and concentration of the purified RNA were assessed using a high-sensitivity Agilent 2100 bioanalyzer (Agilent, Santa Clara, CA, USA) with Agilent RNA 6000 Nano kit under manufacturer’s recommendation.

### 2.3. Total RNA Library Construction and Sequencing

100 ng of total RNA of each sample was prepared for sequencing according to the protocols supplied with the Illumina TruSeq RNA Sample Prep Kit-v2, Set A and B (Illumina, San Diego, CA, USA), with jumping the steps for the enrichment of mRNA or PolyA selection. The barcodes targeted metatranscriptomic cDNA libraries were detected using the DNA 1000 kit (Agilent Technologies) and quantified using the KAPA SYBR Fast qPCR kit for illumina Technologies (Kapa Biosystems, Boston, MA, USA). All the sample libraries were multiplexed into one pool after being normalized based on the qPCR results. All the barcoded fragments were sequenced in the way of paired end (2 × 300 bp) sequencing on all Illumina MiSeq platform using a MiSeq Reagent v3 600 cycle kit (Illumina, San Diego, CA, USA).

### 2.4. Sequences Analysis of Microbiome

The output sequence profiles of the MiSeq platform were analyzed using a pipeline developed in house to obtain a snapshot of the microbial community of the rumen and a comparison file of community composition of the two treatments. A merged read/sample was generated after blasting and assembly paired-end reads (R1&R2) based on the barcodes, from which, rRNA was identified using the rRNA-HMM tool [35] of the Rapid Analysis of Multiple Metagenomes with a Clustering and Annotation Pipeline (RAMMCAP) [36], and then separated into LSU (23S or 28S) and SSU (16S or 18S) rRNA profiles of each taxonomic domains, including archaea, eukaryotes, and bacteria. For subsequent steps, 20,000, 20,000, and 2000 sequences from SSU rRNA profiles of bacteria, eukaryote and archaea were respectively randomly subsampled using the fasta-subsample tool in the MEME 4.10.2 [37] toolkit for furtherly taxonomical analysis. The subsampled SSU rRNA reads of eukaryote and archaeal were binned based on the best hits after searched against the SILVA SSURef-111 database using a threshold e-value of 1e-5 using BLASTN [38]. While the bacterial SSU sequences were binned using the “classify.seqs” command of MOTHUR 1.33.1 [39] into operational taxonomic units (OTUs), for which, the SSURef-108 gene and the SSURef-108b taxonomy databases were used as reference. The binned comparison file was then parsed for furtherly analysis.

### 2.5. Mining of Carbohydrate-Active Enzymes for Muskoxen Fed Triticale Straw

mRNA-Seq libraries and sequence analysis were constructed according to Meng Qi’s method [19]. A metagenomic analyzer, MEGAN [40] was used for function-based taxonomy binning combining the trimmed down non-redundant muskoxen amino acid database (NRMO), which built by our lab [19]. The identified coding regions for GH and carbohydrate-binding modules (CBMs) were further examined using HMMER hmmsearch with Pfam hidden Markov models (HMMs) as previous description [41]. GHs and CBMs were named using carbohydrate-active enzymes (CAZymes) nomenclature. When a Pfam HMM for a given GH either did not exist or did not correspond to the catalytic module, BLAST searches against the data set were performed using regions of sequences listed at the Carbohydrate-Active Enzymes database. In addition, the coding sequences predicted as GHs were further characterized bysimilarity search against NCBI’s protein data bank and non-redundant (nr) database using the BLASTP algorithm [42]. The output obtained were analyzed manually for determining the diversity and abundance of the various CAZymes classes: GHs, CBMs, carbohydrate esterases (CEs), glycosyl transferases (GTs), and polysaccharide lyases (PLs) in the rumen metagenomes.

### 2.6. Statistical Analysis

Annotation information of all randomly picked reads was profiled and a comparison file was presented for all 14 samples. Relative expression values for a taxonomic phylotype in a sample was defined as the ratio of the number of reads assigned to this phylotype to the number of reads assigned to the corresponding kingdom. Means of relative expression value of each taxonomic phylotype was calculated for each treatment. We examined the effects of forage, stage, and feeding sequence on the diversity and abundance of rumen microbial populations, including all bacteria, archaea, fungi and protozoa, under a crossover design. Statistical power of 0.80 (80%) was obtained in this study when the minimally detectable effect size was 1.0 and the significance level was 0.05. A paired t-test was used to determine significant differences for pairwise comparisons between the two forages, two stages, and feeding sequences, statistical significance was noted as Ptr, Pst, and Pco, respectively. Statistically significant differences were further filtered using a relative abundance cutoff of 0.01% for each phylotype and being detected in at least half of the samples for at least one treatment. It meant that a phylotype statistically more abundant in the triticale straw compared with brome hay would be ignored unless it contributed at least 0.01% of the reads assigned to the triticale straw feeding community and was detected in at least half of triticale straw samples. Based on the phylotype-related relative abundances, between-classes principal component analysis (PCA) [30] was taken to analysis the different effects of two forages on the diversity and structure of microbial population using the R package ade4 [43]. A Monte-Carlo test was used to determine significant differences between the two forages [30]. Correlation analysis was employed to visualize the relationships among specific microbial populations and displayed as a heat map of Spearman correlation coefficients.

## 3. Results

### 3.1. Metatranscriptomics Sequence Data Statistics

Under a crossover design, although one animal was removed from trial due to an infection of the horn boss, Illumina next generation sequencing led to the thorough elucidation of rumen microbial diversity and their functional capacity to hydrolyze lignocellulosic biomass in muskoxen rumen under different (roughage) feeding scenario. Metatranscriptomics sequence from 14 biologically independent samples generated about 11,044,327 effective reads having an average of 788,881 effective read. The number of large subunits SunUnitrRNA (LSU) reads was slightly higher than that in small subunit (SSU) rRNA (49.74% vs. 50.19%). In addition, 0.07% of reads could not be classified into LSU rRNA and SSU rRNA clusters (Appendix A).

### 3.2. Total Community Structure and Diversity

The results of β-diversity analysis showed that different feeding stages and feeding order had no significant microbial diversity, while forage type result in a notably difference in rumen microbiome structure and composition (Figure 1A). Specifically, the SSU rRNA reads identified by rRNA-HMM were separated into bacteria, archaea, and eukaryotes. No significant difference for relative abundance of total genotypes of either bacterium or eukaryote in muskoxen rumen fed triticale straw and brome hay (Figure 1B,C; *P* > 0.05). The relative abundance of the archaea was significant higher in triticale straw feeding group than that of brome hay (Figure 1D; *P* = 0.037).

As showed in Figure 1E, forage scenario had no effect on the prokaryotic and eukaryotic population at the phylum level. However, at the genus level, difference of either prokaryotes or eukaryotes between the two forages was visualized and Monte-Carlo test indicated the difference was significant (*P* < 0.05). Comparison was also performed at the genus level for bacteria, archaea, fungi, and protozoa separately (Figure 1F). It indicated that at the genus level, each of the four groups was strikingly distinct between the muskoxen rumen fed either triticale straw or brome hay, as determined by a Monte-Carlo test (n = 999) in all four cases (*P* < 0.05).

### 3.3. Highly Abundant Microbes in Muskoxen Rumen

In muskoxen rumen fed either triticale straw or brome hay, 15 phyla of bacteria were detected, and the most abundant bacteria was Firmicutes, followed by Bacteroidetes, Spirochaetes, Fibrobacteres, Proteobacteria, Actinobacteria (Figure 2A). At the genus level, 73 phylotypes were observed (Figure 2B). *Treponema* and *Fibrobacter* contributed most to their respective phyla, whereas the representations of Firmicutes and Bacteroidetes were more diverse. Major contributions of Firmicutes came from *RFN8-YE57*, *IS_C_leptum_sporosph*, *Ruminococcus*, *vadinHA42* and *Butyrivibrio_fibrisolvens*, and major contributions of Bacteroidetes came from *Prevotella*, *Prevotellaceae_uncultured*, *RC9_gut_group*. Most of the major phylotypes, in particular at the higher taxonomic levels, were of high consistency across environments. 

Regarding archaea, five genera of archaea, i.e., Methanomicrobium, Methanobrevibacter, Methanospaera, Methanonacterium, and Thermoplasmatales were observed (Figure 3A). All organisms are methanogenic archaea and fall into the phylum Euryarchaeota. The most representations of archaea communities were great diverse between the two forages. In the muskoxen rumen fed triticale straw, major contributions came from Methanobrevibacter (36.62%), Methanomicrobium (33.34%), and Thermoplasmatales (14.26%); while in brome hay, major contributions came from Methanobrevibacter (52.65%) and Thermoplasmatales (32.19%), and the abundance of Methanomicrobium was dramatically decreased (1.97%).

As illustrated in Figure 3B,C, the majority of eukaryotic sequences were derived from fungi and protozoa, the relative abundance of fungi was lower in triticale straw than in brome hay, while the relative abundance of Protozoa was higher in triticale straw than in brome hay. In addition, anaerobic fungi observed included *Rhizophydium* and *Spizellomyces*, which belonged to phylum Chytridiomycota, three genera *Neocallimastix*, *Cyllamyces*, and *Piromyces* were included and Neocallimastix and Cyllamyces were the major contributions for the two forages. Six genera of Entodiniomorphid protozoa in the family of *Ophryoscolecidae* (*Entodinium*, *Polyplastron*, *Diplodinium*, *Eudiplodinium*, *Epidinium*, and an uncultured group), two genera of holotrich protozoa in the family of *Isotrichidae* (*Isotricha* and *Dasytricha*), and an uncultured group in the subclass of Trichostomatia were observed in this study. Entodiniomorphid protozoa *Entodinium* and *Polyplastron* were predominant at the genus level for the two forages.

### 3.4. Unfolding Microbial Diversity in Response to Dietary Changes and Ruminant Animal

As shown in Table 1, the selectively enriched bacterial phylotypes mainly distributed in the groups of Bacteroidales, Clostridiales, Cytophagaceae, Chloroflexi, Thermoactinomycetaceae, and four classes of Proteobacteria, as well as two classes of phylum Actinobacteria. The taxa most strongly selected by triticale straw included *Fervidicella*, *Desulfoluna*, *Thermoactinomycetaceae*, *Seinonella*, *Nonlabens*, *SP3-e02-2*, *Betaproteobacteria*, *Desulfobacteraceae*, *IS_C_leptum_sporosph*, *Thermobrachium*, and *Anaerovirgula*. *Selenomonas* was only detected in brome hay, so it was strongly selected by brome hay. Besides *Selenomonas*, other bacteria taxa such as *Rhodospirillales*, *Rhodospirillaceae*, *Roseburia*, *Quinella*, and *Xylanibacter* were strongly selected by brome hay. The selection of fungi by forages included the Cyllamyces and Spizellomyces, which were selected by triticale straw and brome hay, respectively (Table 1). Regarding protozoa, *Entodinium* was strongly selected by triticale straw, and both *Dasytricha* and *Diplodinium* were selected by brome hay strongly and moderately, respectively. The selection of Archaea by forages included *Methanomicrobium*, *Thermoplasmatales*, and *Methanobrevibacter* that moderately selected by triticale straw and brome hay, respectively.

Concerning the correlation analysis between forage and microflora, in triticale straw, there was a significant positive correlation among Proteobacteria (including *Alphaproteobacteria*, *Betaproteobacteria*, *Deltaproteobacteria*, and *Gammaproteobacteria*), Chlamydiae, Synergistetes (including *Synergistia*) CandidatedivisionOD1, Cytophagia, Actinobacteria, Flavobacteria, and Bacilli (named as FTs1_1). Similarly, Firmicutes such as *Clostridia*, *Bacteroidia*, *Coriobacteria*, *Archaea*, *Eukaryotes* also were significantly related (named as FTs1_2). Besides, there was a significant negative correlation between FTs1_1 and FTs1_2, for instance, Firmicutes/Proteobacteria (r = −0.85), Archaea/Proteobacteria (r = −0.97), Archaea/ Cytophagia (r = −0.95) and Eukaryota/Actinobacteria (r = −0.89) (Figure 4A). In brome hay, an apparently positive correlation was notice among Proteobacteria including Deltaproteobacteria, Chlamydiae, Chlamydiae, CandidatedivisionOD1, Actinobacteria, Flavobacteria, and Bacilli that was named as FBh1_1 (Figure 4B).

### 3.5. CAZymes Profile of Muskoxen Rumen Microbiome

The total number of hits obtained for CAZymes classes were 2131–2886 corresponding to 159–184 families. Out of the total CAZymes, GHs were the most abundant class (45.87% to 51.01%) followed by CBMs (20.41% to 22.63%), CEs (10.09% to 18.70%), GTs (5.70% to 12.26%), and PLs (1.97% to 3.07%) (Figure 5A). Further, as showed in Figure 5B and Figure 6, GHs were the major CAZymes involved in cellulose digestion (GH5, GH6, GH9, GH45, GH48, GH94 and GH124). Among cellulases, the rumen microbes contributing to GHs majorly included *Ruminococcus* (70.8 ± 16.6), *Piromyces* (58.8 ± 8.1), *Neocallimastix* (36.5 ± 8.6), *Fibrobacter* (32.5 ± 6.0), *Epidinium* (19.5 ± 7.0), and *Polyplastron* (18.0 ± 3.6). Abundance of major genes coding for hemicellulose digestion was higher than cellulose, and included GH8, GH10, GH11, GH16, GH26, GH30, GH51, GH74, GH114 and GH115. These enzymes were related with *Fibrobacter* (49.7 ± 6.5), *Ruminococcus* (29.0 ± 3.9), *Clostridium* (14.2 ± 3.5), *Piromyces* (13.2 ± 5.2). Enzymes dedicated for hydrolysis of oligosaccharides (GH1, GH2, GH3, GH31, GH32, GH33, GH42, and GH109) were associated with *Prevotella* (24.0 ± 11.5), *Piromyces* (19.2 ± 4.3), *Butyrivibrio* (14.5 ± 3.4), *Bacteroides* (14.0 ± 2.5), *Cellulosilyticum* (12.3 ± 3.0), and *Fibrobacter* (11.0 ± 4.1). In addition, starch degrading enzymes encoded by GH13, GH77, and GH119, which were concerned with *Oxytricha* (26.5 ± 7.4) and *Eudiplodinium* (11.3 ± 3.8).

## 4. Discussion

Ruminants utilize carbohydrate-rich agricultural waste residues such as triticale straw as substantial energy resources. The host itself is unable to synthesize any enzymes required for deconstructing the plant biomass and primarily depend on rumen microbiota to liberate the energy from plant polysaccharides in the form of carbohydrates and sugars. Muskox, as the biggest herbivore in the High Arctic, has been enduring the austere arctic nutritional conditions and has evolved to ingest and digest scarce and high lignified forages to support the growth and reproduce, which could be an outstanding animal to characterize the alteration in rumen microbiome with forage. The SSU rRNA reads identified by rRNA-HMM were separated into bacteria, archaea, and eukaryotes. Of note, the relative abundance of eukaryotic population was proximate to that of bacteria in solid digesta of muskoxen rumen at the transcription level regardless feeding scenario, which was more different than the proportion estimated by traditional technologies [44,45], indicating that eukaryotic population performed high activity responding to the environment. This phenomenon was also observed in cattle (Elekwachi et al., submitted) and in lactating dairy cow [31] that revealed through the same approach, although they were less abundant than in muskoxen rumen. Comtet-Marre et al. revealed that the *Intranmacronulceata* of protozoa actively expressed 542 folds compared with its low relative abundance on a DNA basis [31]. It indicated that the roles of eukaryotic community in the degradation has been beyond our previous understanding.

For comparing the composition and structure of microbial community in muskoxen rumen fed triticale straw or brome hay, our data showed that feeding scenario had no effect on the prokaryotic and eukaryotic population at the phylum level. However, at the genus level, difference of either prokaryotes or eukaryotes between the two forages was visualized and Monte-Carlo test indicated the difference was significant, implying that selection of rumen microbiomes was largely forage specific at the genus level. The difference in the content of NDF, ADF and cellulose between the two forages probably contributed to this selection. In addition, Statistical student t -test under a crossover design showed no significant difference for relative abundance of total genotypes of either bacteria or eukaryota in muskoxen rumen fed triticale straw and brome hay, while archaea were striking affected by forages in muskoxen rumen, and was higher in triticale straw than brome hay, indicated that there was correlation between the methane emission and quality of diets in ruminant, more methane producing in lower quality diet [46].

Concerning the relationship between forages and microbial community, it was established that the rumen flora is dynamic and known to adapt to changes in the host diet [2,47,48]. Recently, cellulolytic *Ruminococcus* spp., *Fibrobacter* spp., and *Cytophaga* spp. clearly played important roles in fiber degradation [8,49,50,51,52,53,54,55]. In the present study, there was no difference in the relative abundance of *Fibrobacter* between triticale straw and brome hay. However, diverse genera in families of *Ruminococcaceae* and *Cytophagaceae* had selectively enriched by triticale straw. *Prevotellaceae*, *Veillonellaceae*, *Coriobacteriaceae*, and *Succinivibrionaceae* were selectively enriched by brome hay. Those groups seemed to play pivotal roles in the degradation of corresponding forage and most of those phylotypes had been only detected in the rumen of Angus heifer fed forage other than high grain [21]. In addition, some species of non-cellulolytic families were thought to be involved in the fermenting of oligosaccharides, starch, sugar to produce formate, acetate, lactate, propionate, butyrate, and other volatile fatty acid that provide most of energy for host, i.e., *Prevotellaceae*, *Rikenellaceae*, *Peptococcaceae*, *Veillonellaceae*, *Succinivibrionaceae*, and *Lachnospiraceae* [11,56,57,58]. In this study, *Prevotellaceae* and its two genera (*Prevotella* and *Xylanibacter*), genus of hoa5-07d05_gut_group of *Rikenellaceae*, *Veillonellaceae* and its genera (*Quinella* and *Selenomonas*), some genera of *Lachnospiraceae* (such as *Acetivibrio_ethanolgignens*, *IS_Eub_rum_Coprococcus_A2_166*, *Pseudobutyrivibrio*, *RC25*, and *Roseburia*), and Succinivibrionaceae were selectively enriched by brome hay. Similarly, *Rikenellaceae* and its genera (*RC9_gut_group* and *SP3-e08*), *Peptococcaceae*, *Propionispira* of *Veillonellaceae*, two genera of *Lachnospiraceae* (*Parasporobacterium* and *Sporobacterium*) were selectively enriched by triticale straw. These data indicated that triticale straw and brome hay could share with homologous non-cellulolytic families to ferment of oligosaccharides, starch, sugar for energy production in muskoxen.

Of note, the accumulating of electron and hydrogen generated in the fermentation would inhibit hydrogenase activity and the oxidation of sugar and would be transferred to acceptors other than oxygen under the anaerobic conditions in rumen. Acetogens would dispose hydrogen with carbon dioxide to produce acetate by the acetyl-CoA pathway [46]. This is an economic approach because it not only disposes the hydrogen, but also recycles the carbon dioxide. Propionogens would sink hydrogen with acrylate and fumarate to produce propionate through propionogenesis [46]. Those two pathways could generate VFA, the main energy source for the host [46]. However, the well-known acetogens, genera of *Acetonema* [59] and *Acetitomaculum* [12], did not show difference between animals fed either triticale straw or brome hay, and *Propionispira* [60] was selectively enriched by triticale straw, which were inconsistent with the things that the produce of acetate or propionate is related to biomass of the fermentable components of forages [46] and brome hay contains more fermentable components than triticale straw. Therefore, there should be other acetogens and propionogens in muskoxen rumen and further experiments should be conducted.

Alternatively, sulfate was another acceptor to sink electron and hydrogen to bioconvert to sulphide by sulfur reducing bacteria [61]. Muskoxen had high intake and digestibility of sulfate in autumn [62]. It indicated that sulfur compounds played important roles in electron transferring and microbiota regulating in muskoxen rumen. Deltaproteobacteria was a class of sulfur-reducing bacteria that can provide oxidizing force through reducing oxidized sulphate to sulphide [61]. All the phylotypes of this class detected in this study, including its derived orders, families, and genera (*Desulfobacterales*, *Desulfobacteraceae*, *Desulfoluna*, *Desulfovibrionales*, *Desulfovibrionaceae*, and *Desulfovibrio*), were selectively enriched by triticale straw. Coupling with the reduction of oxidized sulphate to sulphide, organic moleculars were oxidized and energy was transferred. *Sporobacterium* spp. were identified as possessing the ability to degrade aromatic compound [63,64] and were also selectively enriched by triticale straw. Those psionics could be pivotal in the degradation of aryl ether bonds linking the p-hydroxycinnamates and lignin, as well as the C-C bonds. In an anaerobic environment of rumen, those metabolic capabilities could be supported by the activity of organisms of Deltaproteobacteria. The metabolic synergy performed by those two groups was pivotal for the degradation of the triticale straw. Besides the taxa of Deltaproteobacteria, other phylotypes known to be involved in electron and energy transferring, such as Chloroflexi [65,66], some genus of *Clostridiaceae* (such as *anaerovigula* and *Fervidicella* [9]), *Geosporobacter* [10,67] and *Thermobrachium* [68] were also detected in this study and all were selectively enriched by triticale straw. It indicated that the organisms of those groups played pivotal roles in the transferring of electron and energy in the degradation of triticale straw. Given that triticale straw contained more components that not easy to break down than brome hay, the degradation of triticale straw was harder and more taxa involved, evidenced by the results of the microflora action model in this study.

Methanogens would sink hydrogen with carbon dioxide to produce methane through methanogenesis [46]. Although this pathway prevented the accumulation of reducing equivalents, it was also considered to be sub-economic [46]. Even so, the metabolic activity of methanogens was proved to enhanced the degradation of forages [69]. The supplement of *Methanobrevibacter sp* and *Ruminococcus flavefaciens* to new born lambs increased the population of *R. flavefaciens*, the straw degradation, the activity of some glycoside and polysaccharide hydrolases of the adherent microbial populations and the concentration of acetate in ruminal contents [69]. Thus, there was a trade-off for the proportion of methanogens of the entire rumen microbial population. For the basis of efficient degradation and utilization of lignin enriched forages, it was supposed that the proportion of methanogens in the population of entire community in muskoxen rumen was approximate to ideal. Further, in this study, the selection of methanogens in muskoxen rumen was forage specific. The major archaeal *Methanobrevibacter* and *Thermoplasmatales* were selectively enriched by brome hay, while the *Methanomicrobium* was strongly selectively enriched by triticale straw, which contributed to the difference in the total abundance of archaea between the two forages. In rumen, *Methanobrevibacter* spp. metabolically preferred to the CO_2_ or formate, and H_2_ [70] that produced through the succinate-propionate pathway [71,72]. While *Methanomicrobium* spp. preferred to use acetate as substrate [73]. It indicated that succinate-propionate pathway was the main pathway to produce carbon dioxide in the degradation of brome hay and the contribution of acetogenesis pathway in the degradation of triticale straw was dramatically increased compared to brome hay, although further experiments would be carried out for the characterization of effects of trial stages and feed sequences. This was also evidenced by that the ratio of acetate to propionate was related to biomass of the fermentable components of forages [46] and triticale straw contained less fermentable biomass than brome hay.

With regard to fungi and protozoa, most of the recognized so far anaerobic fungi in ruminant is that they could produce a wide range of high cellulolytic and xylanolytic enzymes to degrade the high-fiber feed particles, which were found to associated with the rhizomycellium and excreted into the surrounding microenvironment, and were known to preferentially adhere to the lignin rich regions and acted as the primary invader to initiate the degradation of fibrous feed particles [74]. *Cyllamyces* (the family *Neocallimastigaceae*) was related to fiber degradation [75], which probably took part in the degradation by rupturing the colonized tissues with bulbous holdfasts [76]. For this, *Cyllamyces* was strongly selected by triticale straw in this study. *Spizellomyces* is in the phylum Chytridiomycota, which is essentially ubiquitous zoospore-producing fungi. Like all chytrids, *Spizellomyces* might be related to the degradation cellulose degradation through producing GH [77]. The specific selection by brome hay suggested that *Spizellomyces* probably was important for the degradation of triticale straw. As far as protozoa is concerned, in addition to degrade the plant cell walls and cell wall fragments by the excreted extracellular enzymes, protozoa also acted as predators of bacteria and fungi, and were responsible for the protein turnover in rumen. Particularly, the small *Entodinium spp* contributed to much of the bacterial predation and protein turnover [78]. In the present study, protozoa accounted a large proportion of the entire microbial population and *Entodinium* contributed the majority in muskoxen rumen. The outcomes of the selection of protozoa by forages showed that *Dasytricha* and *Diplodinium* were selected by brome hay, while *Entodinium* was enriched by triticale straw. Study that focused on ruminal fermentation and microbial community responding to four typical subtropical forages in vitro indicated that the most abundance of *Entodinium* was observed with cassava residues when compared to corn straw silage and elephant grass [79]. The alteration in protozoa may be highly associated with the requirement for high efficiency of utilizing energy of muskoxen.

Rumen microbes harbor complex array of enzymes which work synergistically for plant fiber degradation, including GHs, CBMs, CEs, GTs, PLs, etc. Of various CAZyme family, GHs are the most abundant and diverse group of enzymes accountable for hydrolyzing glycosidic bonds of carbohydrates in plant polysaccharide [80]. Date from buffalo and cattle rumen showed that the presence of 72 and 78 distinct families of GHs, respectively, to hydrolyze the various plant cell wall constituents [33,81]. In this study, the notably enormous enzymes categorized into GH family in the rumen metagenome of muskoxen (159 to 189 families) indicated a more complex process of lignocellulose breakdown compared to others. In particular, a higher proportion of genes from GH48, GH5, GH45, and GH9 families encoding endoglucanase were present in the present study. Previous analogous studies have reported that GH48 family that one of the components in most of the cellulosome system was found to be the most abundant enzyme subunit in cellulosome-producing bacteria [82]. In addition to endoglucanase activity, GH5 family proteins are also reported to exhibit endo-xylanase mannanase and exoglucanase activity [83]. The higher abundance of genes encoding putative GH5 family in this study indicates that rumen microbiome of muskoxen may comprise a unique approach for plant polycarbohydrate deconstruction. Moreover, in line with the other studies [84], we also noticed that the abundance of major genes coding for hemicellulose digestion was higher than cellulose. This was expected since the diversity of side chains in hemicellulose requires a diverse pool of hydrolytic enzymes for its degradation. Compared with the muskoxen mixed fed triticale straw and brome grass hay [19], the muskoxen fed triticale straw exhibited lower GH10 abundance (4.69% vs. 6.96%) and higher levels of GH8 (2.07% vs. 0.59%) and GH11 (5.81% vs.4.41%), indicating that the GH8 and GH11 exerted critical role in hemicellulose degradation for muskoxen. Apart from GHs, other CAZyme classes including CBMs, GTs and PLs, are involved in hydrolysis of cellulose, hemicelluloses, xylan and pectin [85]. CBMs which do not exhibit the enzymatic activity of their own but helps in binding CAZymes to polysaccharide, thus potentiating their activities [86], accounted for 20.41% to 22.63% of CAZymes in the current study. CEs play key role in breaking the ester bonds between lignin and carbohydrates in the rumen [85], and were found to be the third most profuse CAZy family (10.09% to 18.70% of total CAZyme). Nearly 5.70% to 12.26% of the CAZymes were found to have similarity with class CEs that facilitate the action of catalyzing the breakage of glycosidic linkage [87].

Rumen microbes like *Fibrobacteraceae*, *Clostridiaceae*, *Ruminococcaceae*, *Prevotellaceae*, *Bacteroidaceae*, and *Lachnospiraceae* have been reported to be the major fiber degrading microbes in muskoxen [19]. Accordingly, *Prevotella* has been identified closely related with GHs involved in the degradation of oligosaccharides and hemicelluloses [88]. *Ruminococcus* and *Fibrobacter* affiliated to the majority of the putative cellulases belonged to GH families, especially GH5, GH9, GH45, and GH48 [32]. Perhaps therefore, *Ruminococcus* is proficient in the deconstruction of cellulose and hemicellulose [89]. In the present study, our date showed that major fiber degrading rumen microbes like *Bacteroides*, *Butyrivibrio*, *Cellulosilyticum*, *Clostridium*, *Fibrobacter*, *Prevotella*, *Ruminococcus*, etc. contributed to GHs. It is well-known that cellulose and hemicellulose in plant biomass are considered as one of the prevalent renewable resources of fermentable sugars that could be utilized further in numerous industrial processes, like ethanol generation. Therefore, the distinct microbial reservoir in muskoxen reflects its efficiency for the deconstruction of plant biomass. Nonetheless, further research is required to explore the rumen microbiome and the way it carries out fiber deconstruction

## 5. Conclusions

Ruminants harbor a vast and diverse microbial community that functions in the utilization of fibrous feedstuffs. In the present study, comparative metatranscriptomics approach was employed to portray the rumen microbial diversity and define the relationship between fiber digestion and microbial communities of muskoxen fed on either triticale straw or brome hay. The global effects of triticale straw and brome hay on the prokaryotic and eukaryotic population, and of the lower clade (bacteria, archaea, fungi, and protozoa) were observed at the lower taxonomic level of genus, indicating more detailed selection in the degradation of two forages original from the same gramineae. Further analysis revealed that triticale straw with higher content of NDF, ADF, cellulose selectively enriched more lignocellulolytic taxa and electron transferring taxa, while brome hay with higher N content selectively enriched more families and genera for degradable substrates-digesting. Further analysis revealed the existence of enormously diverse CAZyme belonging to classes: GHs, CBMs, CEs, GTs, and PLs, with the dominance of GHs. Of the total GHs identified were found to be responsible for direct plant polysaccharide deconstruction, especially cellulose and hemicellulose. The presence of higher bacterial population and its close connection with CAZyme concludes that the rumen of muskoxen as an important converser of plant biomass to high-value products.

## Figures and Tables

**Figure 1 microorganisms-10-00071-f001:**
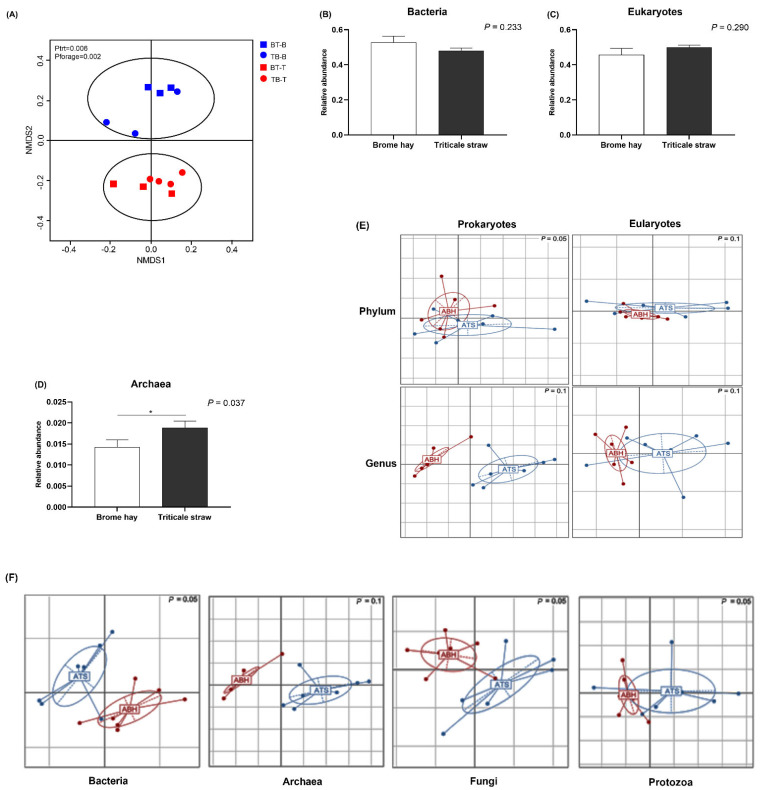
The structure of microbial community response to forages scenario. (**A**) the effect of feeding stages, feeding order, and forages scenario on the structure microbiome was analyzed. The composition of microbiota in (**B**) bacteria, (**C**) eukaryotes, and (**D**) archaea of muskoxen rumen. Between-classes PCA revealing the differences of microbial population structure of solid-phase rumen contents from muskoxen in two groups crossover fed either triticale straw (blue circle noted) or brome hay (red circle noted) in term of (**C**) prokaryotes and eukaryotes at the levels of phylum and genus, as well as (**D**) in population structure of bacteria, archaea, fungi, and protozoa at the genus level. The statistical significance of the between-classes PCA, as determined by a Monte-Carlo test (n = 999), gave *P* ≤ 0.05 in all four cases. The letters of ATS and ABH refer to the muskoxen fed triticale straw in autumn and brome hay in autumn, respectively. * *P* < 0.05.

**Figure 2 microorganisms-10-00071-f002:**
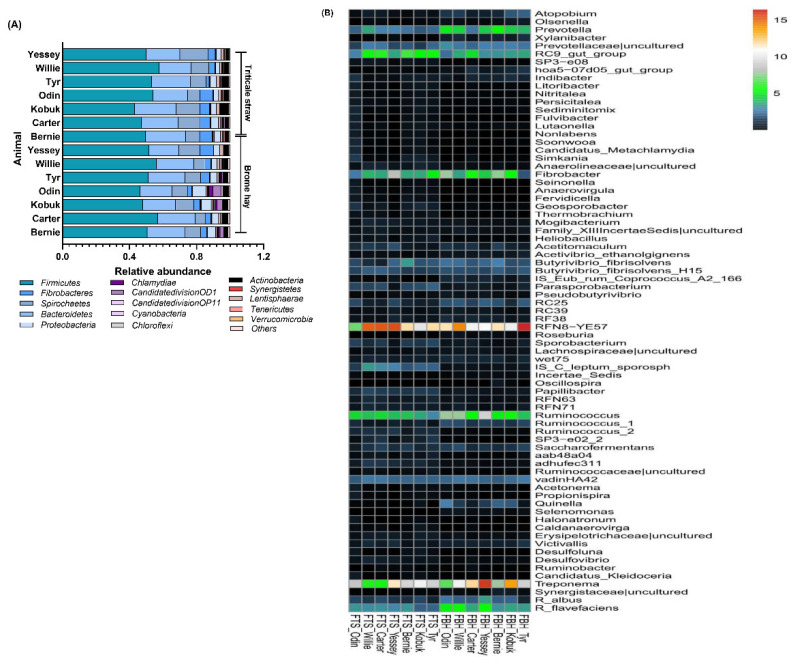
Bacterial phylotypes at the level of (**A**) phylum and (**B**) species in solid phase rumen contents from 7 muskoxen in two groups crossover fed either triticale straw (TS noted) or brome hay (BH noted).

**Figure 3 microorganisms-10-00071-f003:**
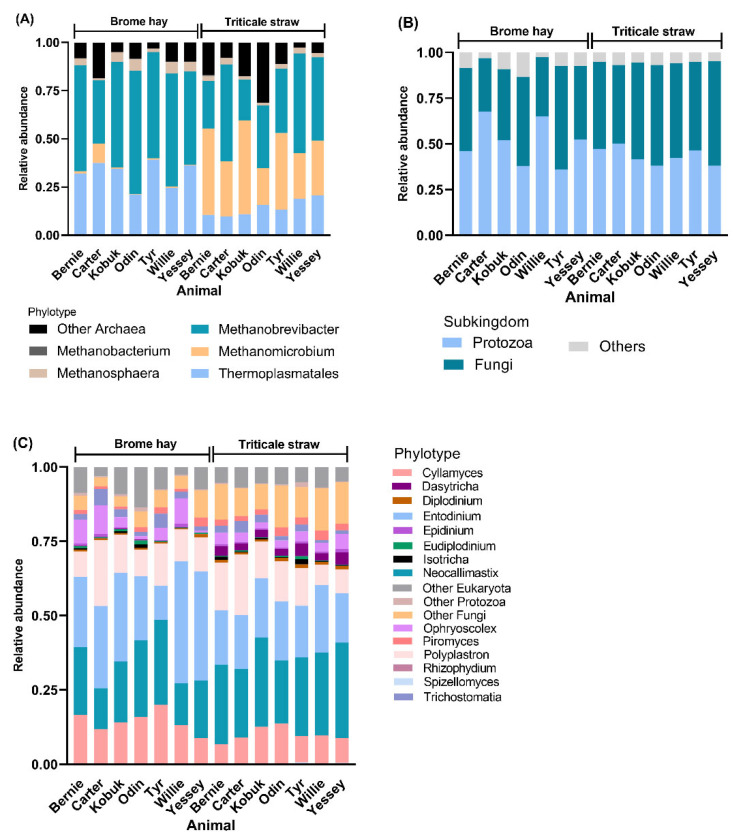
Contribution of archaea and eukaryotic in solid phase rumen contents from muskoxen in two groups crossover fed either triticale straw (TS noted) or brome hay (BH noted); (**A**) The relative abundance of archaeal phylotypes of the muskoxen rumen on either triticale straw or brome hay during fall, and (**B**) the composition eukaryotes of the muskoxen rumen, as well as (**C**) the distribution of eukaryotic community at the genus level in solid phase rumen contents.

**Figure 4 microorganisms-10-00071-f004:**
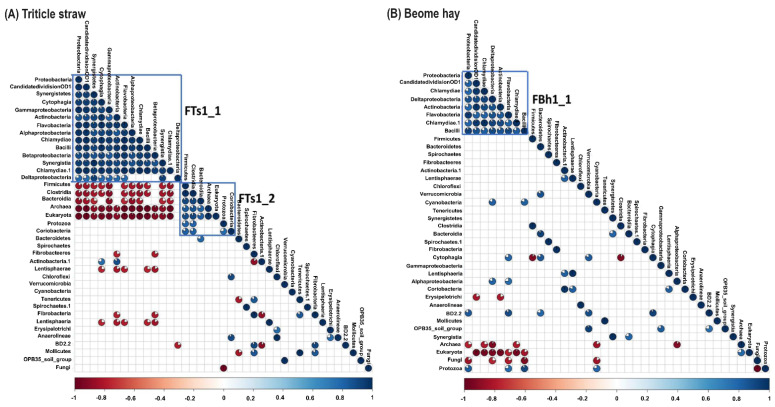
Correlation matrixes of all the phylotypes at the ranks of king, phyla, and class in the rumen solid samples of muskoxen on (**A**) triticale straw and (**B**) brome hay during fall. Squares denotes the phylotypes which can be detected at least 75% analyzed samples. Blank squares denote *P* > 0.05, while the colored pie filled squares denote *P* ≤ 0.05; both the sector size and the colors denote the correlations, and the colors are presented in the scale bar denoting the nature of the correlation with 1 indicating perfect positive correlation (dark blue) and 1 indicating perfect negative correlation (dark red) between two microbial populations.

**Figure 5 microorganisms-10-00071-f005:**
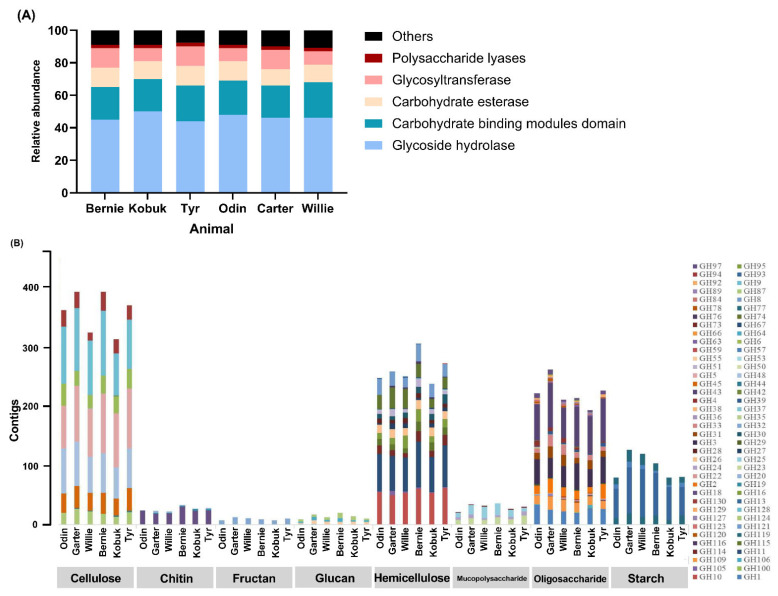
Mining of carbohydrate--active enzymes (CAZymes) for muskoxen fed triticale straw. (**A**) Comparison of predicted CAZymes classes: glycoside hydrolases (GHs), carbohydrate-binding modules (CBMs), carbohydrate esterases (CEs), glycosyl transferases (GTs), and polysaccharide lyases (PLs) in the rumen metagenomes. (**B**) Distribution of predicted GHs family responsible for cellulose, chitin, fructan, glucan, hemicellulose, mucopolysaccharide, oligosaccharide, and starch hydrolysis in rumen metagenome samples based on CAZy database.

**Figure 6 microorganisms-10-00071-f006:**
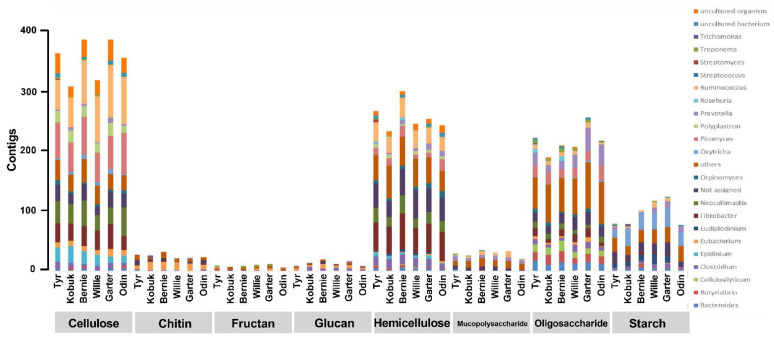
Putative microorganisms involved in the breakdown of cellulose, chitin, fructan, glucan, hemicellulose, mucopolysaccharide, oligosaccharide, and starch hydrolysis. Substrates degrading involved microorganisms were determined by the known activities of their original glycoside hydrolases. Phylotypes which were binned for less than ten contigs of glycoside hydrolases were included in the group of “others”, and the glycoside hydrolases without binned phylotypes were included in the group of “Not assigned”.

**Table 1 microorganisms-10-00071-t001:** The phylotypes selectively enriched by forages.

Item		Triticale Straw	Brome Hay
Bacteria	Phylum	Proteobacteria, CandidatedivisionOD1, Chloroflexi, and Chlamydiae	Synergistetes, Tenericutes, and Cyanobacteria
	Class	Chlamydiae, Deltaproteobacteria, Anaerolineae, Actinobacteria, and Betaproteobacteria	Coriobacteria, BD2-2, Synergistia, and Mollicutes
	Order	Chlamydiales, Micrococcales, Rickettsiales, Anaerolineales, Desulfovibrionales, Bacillales, Burkholderiales, Rhodobacterales, and Thermoanaerobacterales	Coriobacteriales, RF9, Synergistales, Rhodospirillales, and Aeromonadales
	Family	*Ruminococcaceae*, *Rikenellaceae*, *BS11_gut_group*, *Clostridiaceae*, *Simkaniaceae*, *Anaerolineaceae*, *Desulfovibrionaceae*, *Heliobacteriaceae*, *Peptococcaceae*, *Thermoactinomycetaceae*, *Cytophagaceae*, *Family_IIIIncertaeSedis*, *Desulfobacteraceae*, *Flammeovirgaceae*, *Cystobacterineae*, *Flavobacteriaceae*, *and Parachlamydiaceae*	*Prevotellaceae*, *Veillonellaceae*, *Coriobacteriaceae*, *S24-7*, *Synergistaceae*, *Rhodospirillaceae*, *and* *Succinivibrionaceae*
	Genus	*IS_C_leptum_sporosph*, *RC9_gut_group*, *Parasporobacterium*, *Papillibacter*, *Ruminococcus_2*, *SP3-e02_2*, *Sporobacterium*, *adhufec311*, *Geosporobacter*, *Simkania*, *SP3_e08*, *Thermobrachium*, *RFN63*, *Desulfovibrio*, *Anaerolineaceae|uncultured*, *Fervidicella*, *Heliobacillus*, *RFN71*, *aab48a04*, *Anaerovirgula*, *Erysipelotrichaceae|uncultured*, *Ruminococcaceae|uncultured*, *Seinonella*, *Nonlabens*, *Mogibacterium*, *Desulfoluna*, *Persicitalea*, *Caldanaerovirga*, *Fulvibacter*, *Candidatus_Kleidoceria*, *Sediminitomix*, *Candidatus_Metachlamydia*, *and Propionispira*	*Ruminococcus*, *Prevotella*, *Quinella*, *Atopobium*, *RC25*, *Ruminococcus_1*, *Xylanibacter*, *IS_Eub_rum_Coprococcus_A2_166*, *hoa5-07d05_gut_group*, *Selenomonas*, *Roseburia*, *Acetivibrio_ethanolgignens*, *RF38*, *Olsenella*, *Incertae_Sedis*, *Synergistaceae|uncultured*, *Pseudobutyrivibrio*, *and wet75*
	Species		*R. flavefaciens*, *and R.albus*
Fungi		*Cyllamyces*	*Spizellomyces*
Protozoa		*Entodinium*	*Dasytricha*, *and Diplodinium*
Archaea		*Methanomicrobium*	*Thermoplasmatales and Methanobrevibacter*

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
