# Peer review of "Characterizing the Alteration in Rumen Microbiome and Carbohydrate-Active Enzymes Profile with Forage of Muskoxen Rumen through Comparative Metatranscriptomics"

_microorganisms, 2021, doi:10.3390/microorganisms10010071_

Round 1
Reviewer 1 Report
The authors have addressed some of my comments. However, some of the comments has not been addressed in the revised manuscript and a reply for the comments has not been provided. Furthermore, line numbers were not available and this makes difficult to make specific comments.
Major comments:
- It is still not clear how the authors considered the different copy number of rRNA genes in different groups. The difference can be really high in particular between prokaryotes and eukaryotes. How did the authors solve this issue? I still think that comparisons such as the ones reported in Figure 1B and related text are not correct. Please, justify this aspect.
- Chemical data are still missing, is thus difficult to link specific microbial populations a particular condition in the rumen.
- A functional analysis on mRNA has been added in the revised version. However, non-coding RNA was not removed and it could be up to 99% of the total RNA (as stated in the introduction, page 3). Is the sampling depth enough if only 1% of the sequences are mRNA? Please, reply and discuss.
- The discussion regarding fungi and protozoa could be implemented. For example, how do the authors discuss the presence of Rhizophydium and Spizellomyces?
Minor comments:
- I would suggest to further improve the quality of the figures, since the fonts in Figure 2 and in Figure 3 are really small. Furthermore, the labels of Figure 2C are partially covered by covered Figure 2B.
- Page 5, results section. Which is the difference between clean reads and effective reads?
- It is not clear how many unclassified taxa were detected and how their abundance was considered in the calculations.
- Page 11. “Concerning the microflora action model”. It is not a model, it is a correlation analysis.
- Page 16. “The non-rhizoidal Cyllamyces probably took part in the degradation by rupturing the colonized tissues with bulbous holdfasts, this mode of degradation was proposed to be important for the degradation of triticale straw” needs a reference.
Author Response
Responses to the reviewers’ comments
Dear Editors and Reviewers:
We are truly grateful to yours and other reviewers’ critical comments and thoughtful suggestions. Based on these comments and suggestions, we have made careful modifications on the original manuscript. All changes made to the text are in red colour. We hope the new manuscript will meet your magazine’s standard. Below you will find our point-by-point responses to the reviewers’ comments/ questions:
Reviewer 1
The authors have addressed some of my comments. However, some of the comments has not been addressed in the revised manuscript and a reply for the comments has not been provided. Furthermore, line numbers were not available and this makes difficult to make specific comments.
Thanks for your comments
Major comments:
1. It is still not clear how the authors considered the different copy number of rRNA genes in different groups. The difference can be really high in particular between prokaryotes and eukaryotes. How did the authors solve this issue? I still think that comparisons such as the ones reported in Figure 1B and related text are not correct. Please, justify this aspect.
Indeed, there is big different between the copy number of prokaryotes and eukaryotes, and it would be more rigorous if added a test for copy number of those two kingdoms, especially for the analysis of highly abundant microbes. Considered the purpose of our paper is to explore the alteration in rumen microbiome and the relationship among the microbes based on the changes of the relative abundance. The comparation between prokaryotes and eukaryotes not included in our manuscript.
In addition, to clearly exhibit the alteration in archaea, bacteria, and eukaryotes, we improved Figure 1B. Thanks.
2. Chemical data are still missing, is thus difficult to link specific microbial populations a condition in the rumen.
We are appreciative of the reviewer’s suggestion. In addition, it will be more convincing if we provide more evidence of chemical data of the rumen, unfortunately, there are some hurdles to us main due to the period of animal experiment. Therefore, the referee’s concern is of importance for our further study, and we will show the results in our next paper. Here, we seek for the reviewer and editor’s tolerance and understanding. Many thanks for your kind help!
3. A functional analysis on mRNA has been added in the revised version. However, non-coding RNA was not removed and it could be up to 99% of the total RNA (as stated in the introduction, page 3). Is the sampling depth enough if only 1% of the sequences are mRNA? Please, reply and discuss.
As for the referee’s concern, the mRNA sequences used for mining of carbohydrate -active enzymes were not collected from the libraries based on total RNA. The mRNA-libraries were conducted according to Meng Qi’s method (2011) and the mRNA were enriched from independent RNA samples using the Illumina mRNA-Seq sample preparation kit.
Qi M, Wang P, O'Toole N, Barboza PS, Ungerfeld E, Leigh MB, Selinger LB, Butler G, Tsang A, McAllister TA, Forster RJ. Snapshot of the eukaryotic gene expression in muskoxen rumen--a metatranscriptomic approach. PLoS One. 2011;6(5): e20521.
4. The discussion regarding fungi and protozoa could be implemented. For example, how do the authors discuss the presence of Rhizophydium and Spizellomyces?
As your suggestion, we implemented the discussion regarding fungi and protozoa. For example, “…… Cyllamyces (the family Neocallimastigaceae) was related to fiber degradation [75], which probably took part in the degradation by rupturing the colonized tissues with bulbous holdfasts [76]. For this, Cyllamyces was strongly selected by triticale straw in this study. Spizellomyces is in the phylum Chytridiomycota, which is essentially ubiquitous zoospore-producing fungi. Like all chytrids, Spizellomyces might be related to the degradation cellulose degradation through producing GH [77]. The specific selection by brome hay suggested that Spizellomyces probably was important for the degradation of triticale straw.…...” and “In the present study, protozoa accounted a large proportion of the entire microbial population and Entodinium contributed the majority in muskoxen rumen. The outcomes of the selection of protozoa by forages showed that Dasytricha and Diplodinium were selected by brome hay, while Entodinium was enriched by triticale straw. Study that focused on ruminal fermentation and microbial community responding to four typical subtropical forages in vitro indicated that the most abundance of Entodinium was observed with cassava residues when compared to corn straw silage and elephant grass [79]. The alteration in protozoa may be highly associated with the requirement for high efficiency of utilizing energy of muskoxen.”.
Minor comments:
1. I would suggest to further improve the quality of the figures, since the fonts in Figure 2 and in Figure 3 are really small. Furthermore, the labels of Figure 2C are partially covered by covered Figure 2B.
Thanks, we improved the quality of all Figures according to your comment.
2. Page 5, results section. Which is the difference between clean reads and effective reads?
Sorry for this confusion. Clean reads and effective reads are the same mean but different expression. Thus, we unified the expression.
3. It is not clear how many unclassified taxa were detected and how their abundance was considered in the calculations.
According to the approach of metatranscriptomics, all the reads were blasted to the whole sequences but not the segment of 16S or 18S rRNA molecular available from the sequence database online, so there were no unclassified taxa at the levels of phylum, class, and order, but exactly, there were eight uncultured taxa including seven genus and one family taxa.
4. Page 11. “Concerning the microflora action model”. It is not a model, it is a correlation analysis.
We replaced " Concerning the microflora action model " with "Concerning the correlation analysis between forage and microflora "
5. Page 16. “The non-rhizoidal Cyllamyces probably took part in the degradation by rupturing the colonized tissues with bulbous holdfasts, this mode of degradation was proposed to be important for the degradation of triticale straw” needs a reference.
Thanks for your suggestions, we added a reference following by the sentence in revision manuscript.

Reviewer 2 Report
The reviewer would like to appreciate author's intensive revision which have elevated their report to a level of receiving attentions from a wide range of readers. Indeed extensive metatranscriptome dataset about Cazymes newly involved in the report is likely to become one distinctive source for future research. Discussion part is still lengthy, but not too redundant to hamper authors consideration. I would like to suggest a few minor corrections on this revision which contains small awkwardness prior to publication:
1) I came to imagine that there is a situation for authors had to follow an experiment guideline issued by a university where no authors was affiliated, and to get permission no from an institute who is, even in some instance, responsible for this study. Could you denote in the main text about the reason for applying such unorthodox permission scheme.
2) There are spacious rooms for correcting nomenclature about taxonomic description about microorganisms. In principle, Italic should be applied for the family (sometimes the genera) levels and the lower. In their ms in some part (particularly in Table 1, and the following few pages) all taxa described in Italic whereas texts in page 8 and 9 were the reverse, all in Roman. Pl correct all parts in a consistent manner.
Author Response
Responses to the reviewers’ comments
Dear Editors and Reviewers:
We are truly grateful to yours and other reviewers’ critical comments and thoughtful suggestions. Based on these comments and suggestions, we have made careful modifications on the original manuscript. All changes made to the text are in red colour. We hope the new manuscript will meet your magazine’s standard. Below you will find our point-by-point responses to the reviewers’ comments/ questions:
Reviewer 2
The reviewer would like to appreciate author's intensive revision which have elevated their report to a level of receiving attentions from a wide range of readers. Indeed, extensive metatranscriptome dataset about Cazymes newly involved in the report is likely to become one distinctive source for future research. Discussion part is still lengthy, but not too redundant to hamper authors consideration. I would like to suggest a few minor corrections on this revision which contains small awkwardness prior to publication:
Thanks for your comments
1) I came to imagine that there is a situation for authors had to follow an experiment guideline issued by a university where no authors were affiliated, and to get permission no from an institute who is, even in some instance, responsible for this study. Could you denote in the main text about the reason for applying such unorthodox permission scheme.
The Correspondence author Robert J. Forster is an Adjunct Professor University of Alaska. Therefore, in this study, all the animals involving were cared in agreement with the protocol of by the Institutional Animal Care and Use Committee at the University of Alaska Fairbanks and fed at the Robert G. White Large Animal Research Station, Fairbanks, AK. (No. 139821)
2) There are spacious rooms for correcting nomenclature about taxonomic description about microorganisms. In principle, Italic should be applied for the family (sometimes the genera) levels and the lower. In their ms in some part (particularly in Table 1, and the following few pages) all taxa described in Italic whereas texts in page 8 and 9 were the reverse, all in Roman. Pl correct all parts in a consistent manner.
We thank the reviewer’s good advice. We have done corresponding revision according to comment in whole manuscript.

Round 2
Reviewer 1 Report
I thank the authors since they have addressed most of my comments.
This manuscript is a resubmission of an earlier submission. The following is a list of the peer review reports and author responses from that submission.
Round 1
Reviewer 1 Report
Comments to the author
In this ms authors assessed rumen microbiome structure of muskoxen living in Arctic. They chose metatranscriptome technology targeting total rRNA molecules which enabled one shot detection of the three kingdoms instead of genomics approach based on the genes. As such I acknowledge that they succeeded to help us extend our knowledge about novel example of rumen microbiome community of ruminant in a particular circumstance. Meanwhile, I still feel reluctant to recommend their ms to this Journal for publication mainly because of their ms preparation in which criticisms are divided into three:
1) Most critical one was authors contributions, the animal experiment including animal feeding and sampling was done at a station in an university where no author was affiliated. It seems very strange to believe that the University endorsed a group that all members were in other institutes to conduct an experiment using animals owned by the University? How was the scheme to be clarified?
2) In Discussion section they should inherently have provided more intelligent discussion about the specificity of rumen microbial structure of this particular animal compared to other ruminants, but actually did a simple comparison only using observed data from two groupings of feed types. It might be of some necessity, but could be much simpler.
3) Really simply I found many errors and awkwardness on their ms, English grammar errors, inconsistent citation style, figures in wrong aspect ratio, for instance, that clearly reflects their jumping off from paying least care prior to submission.
Reviewer 2 Report
Dear authors,
I have read your manuscript with interest. In general the manuscript is easy to read, I have only some minor remarks:
General comment: You use only 7 animals divided into 2 groups. How valid are the difference you found? Did you do a power calculation before hand?
Line60/61: The references Thomas et al and Stiverson et al. should be added to the reference list and shown as numbers
Line102: Fig.1 should be Figure 1A
Line150: Please change furtherly into further
Line184: Please change 788881 into 788,881
Figure3: The font size in Figure 3 is so small that it is very hard to read. Please increase the font size.
Line 292: please change were into was
Figure4: The font size in Figure 3 is so small that it is very hard to read. Please increase the font size.
Reviewer 3 Report
In this manuscript the authors describe the composition of active microorganisms in muskoxen rumen. The animals were fed tow diets (i.e., brome hay and triticale straw) and total RNA was extracted and sequenced. Since total RNA was sequenced (the mRNA enrichment step was not performed) the rRNA sequences were used to estimate the abundance of the prokaryotic (bacteria and archaea) and eukaryotic (fungi and protozoa) populations and differences between the two diets were observed.
The strength of this manuscript is that the authors attempt to the describe the microbial communities focusing on the active populations, which is not possible in the studies in which DNA is sequenced.
One of the main week point of this manuscript is that it is not considered that different microbial species can harbor a different number of copies of the rRNA genes. The difference can be really high in particular between prokaryotes and eukaryotes. For this reason, in my opinion, is not correct to compare the abundance of such populations (prokaryotes, fungi and protozoa) referring to the abundance of rRNA (e.g., Fig. 1C, Fig. 2D, Fig. 2E and related text, lines 318-319).
Another weak point is that no chemical data are reported, is thus difficult to link specific microbial populations a particular condition in the rumen.
Lines from 339 to 426 were used to discuss the prokaryotic community, but only lines from 427 to 448 were used to discuss the eukaryotic community. The discussion regarding fungi and protozoa could be implemented. For example, how do the authors discuss the presence of Rhizophydium and Spizellomyces?
Furthermore, both in the Introduction and in the Conclusions the authors stress that the rumen can be a source of useful enzymes for the conversion of plant biomass (e.g., lines 38-42, lines 462-465). This is true, but is it not consistent with the approach used in the study, since in this study the functional genes were not targeted.
Other comments:
- Figure 1A should be moved in the Materials and Methods section.
- The overall quality of the figures is low, and often is not possible to read the labels
- Lines 181-182. The functional profile was not characterized in this study, since mRNA was not enriched.
- Line 184. Which is the difference between clean reads and effective reads?
- It is not clear how many unclassified taxa were detected and how their abundance was considered in the calculations.
- Lines 286-297. I did not find these correlations in the Materials and Methods section.
- Lines 315-318. In this study a comparison with the traditional technologies was not performed actually.
- Lines 324-325. This statement is true only if referred to the moment when the samples were collected. The colonization of plant biomass by the rumen microorganisms is a complex process, in which different populations change their abundance over time.
- Lines 337-338. Methane emissions were not measured in this study, therefore, this sentence seems a speculation.
- Lines 363-365. Can the authors explain what they mean with this sentence, please?
- Lines 378-401. Accumulation of sulfide in the rumen could be toxic for the animal. This aspect could be discussed.
- Lines 419-423. I would not write “it indicated” but rather “it suggested” since the functional analysis on these pathways was not performed.
- Line 443-444. I do not understand the link of these lines to ones above (lines 441-442).